# GIFT: An ImageJ macro for automated fiber diameter quantification

**Jennifer Huling** * , **Andreas Götz** , **Niels Grabow, Sabine Illner**

Institute for Biomedical Engineering, Rostock University Medical Center, Rostock, Germany

☯ These authors contributed equally to this work.
* Jennifer.huling@uni-rostock.de

**Data Availability Statement:** The GIFT macro, manual with instructions for installation and use, and the SEM image files used in this paper are available at https://github.com/IBMTRostock/GIFT.

## Abstract

This paper details the development and testing of the GIFT macro, which is a freely available program for ImageJ for the automated measurement of fiber diameters in SEM images of electrospun materials. The GIFT macro applies a validated method which distinguishes fiber diameters based on distance frequencies within an image. In this work, we introduce an applied version of the GIFT method which has been designed to be user-friendly while still allowing complete control over the various parameters involved in the image processing steps. The macro quickly processes large data sets and creates results that are reproducible and accurate. The program outputs both raw data and fiber diameter averages, so that the user can quickly assess the results and has the opportunity for further analysis if desired. The GIFT macro was compared directly to other software designed for fiber diameter measurements and was found to have comparable or lower average error, especially when measuring very small fibers, and reduced processing times per image. The macro, detailed instructions for use, and sample images are freely available online (https://github.com/IBMTRostock/GIFT). We believe that the GIFT macro is a valuable new tool for researchers looking to quickly, easily and reliably assess fiber diameters in electrospun materials.

## Introduction

Nonwoven fiber materials made via electrospinning have been used in a variety of applications including filters, sensors and medical products [1]. Electrospinning is a simple and fast technique that is compatible with a wide range of materials [2]. The resulting fibrous structures benefit from the high surface-to-volume ratio and can be tuned to have the desired porosity, fiber size and fiber alignment. Electrospinning is of particular interest in biomedical applications because the structure mimics the fibrous features of native tissue extracellular matrix and is known to enhance cell interactions [1]. Electrospun materials have been researched in medical applications ranging from drug delivery devices to in vitro engineering of whole tissues [3]. Several medical products have made it to commercial production including vascular stent coverings and wounds dressing [4].

In all applications, one characteristic that is critical to the function of electrospun materials is the average fiber diameter. In the case of medical face masks, reducing fibers from micro- to

All other relevant data are within the paper and its Supporting Information files.

**Funding:** This work was supported by the Federal Ministry of Education and Research under the project RESPONSE-"Partnership for Innovation in Implant Technology" (JH and AG). The funders had no role in study design, data collection and analysis, decision to publish, or preparation of the manuscript.

**Competing interests:** The authors have declared that no competing interests exist.

nano-sized in electrospun filters can reduce air resistance and increase surface area for particle capture [5]. Control of fiber diameter can also be used to tailor drug release profiles [6, 7]. Additionally, changes in fiber diameter of biocompatible materials have been shown to effect cell proliferation, differentiation and morphology [8–10]. Therefore, the ability to easily and accurately analyze fiber diameter is critical in the development and production of electrospun materials.

The most common method for the measurement of fiber diameters is via the manual measurement of fibers based on scanning electron microscopy (SEM) images using free tools like ImageJ and Fiji [11, 12]. However, measurement by hand is slow and can be influenced by the individual making the measurements [13]. A more ideal alternative is automated image analysis. Automated analysis could potentially eliminate user bias and allow for high throughput screening for the development of new electrospun materials or rapid quality control of commercial products. There have been many proposed methods and existing tools for automated fiber measurements, but all have weaknesses including limited software accessibility, high learning curve, or sensitivity to image quality or user input [14–17].

Our research group developed and published a new image analysis method for fiber diameter measurement called the General Image Fiber Tool (GIFT) [14]. The GIFT method works by measuring the distances between edges in a given image and determines which of those distances represent fiber diameter based on their frequency, therefore bypassing the need to specifically identify fibers within an image. The method proved to be accurate and faster than manual measurements or semi-automated measurements requiring user input. However, the initial version of GIFT, which was aimed at testing and validating the theory and method, was not adequately streamlined, relying on the manual transfer of data between programs and the use of proprietary software. Therefore, we felt that the positive results justified further development of GIFT into a standalone, user-friendly piece of software that could be made freely available to the research community.

In this work, we present the development of GIFT into a macro for ImageJ. The GIFT macro allows for large sets of SEM images to be processed quickly in a single step, while allowing users easy control over all of the image processing parameters. The GIFT macro is open source and freely available along with in-depth instructions for its installation, use and troubleshooting. We believe that the GIFT macro has the potential to be highly applicable in the electrospinning research community and to manufacturers requiring SEM imaging as a standard inspection method for monitoring micro and nanofiber products. Additionally, we anticipate that the new, accessible and user-friendly version of the method presented here will facilitate its application and use.

## Materials and methods

### Image processing and macro development

The theory behind the GIFT method was developed, tested and validated previously [14]. In the current work, we have adapted the image processing steps to function entirely within ImageJ for optimal ease of use and accessibility. A summary of the main changes and improvements is given in S1 Table in S1 File. Every image processed is first cropped as needed, then a Sobel operator is applied to the 8-bit image to identify the edges, followed by percent-based thresholding. As an improvement from the original GIFT method, the GIFT macro calculates the threshold cutoff value based on a percent of total included pixels, rather than defining the cutoff value by a single constant grayscale value. The threshold is set so that a user-provided percent of dark pixels remains after thresholding and this is calculated for each image individually during processing. Compared to setting a static grayscale value cutoff, this was found to

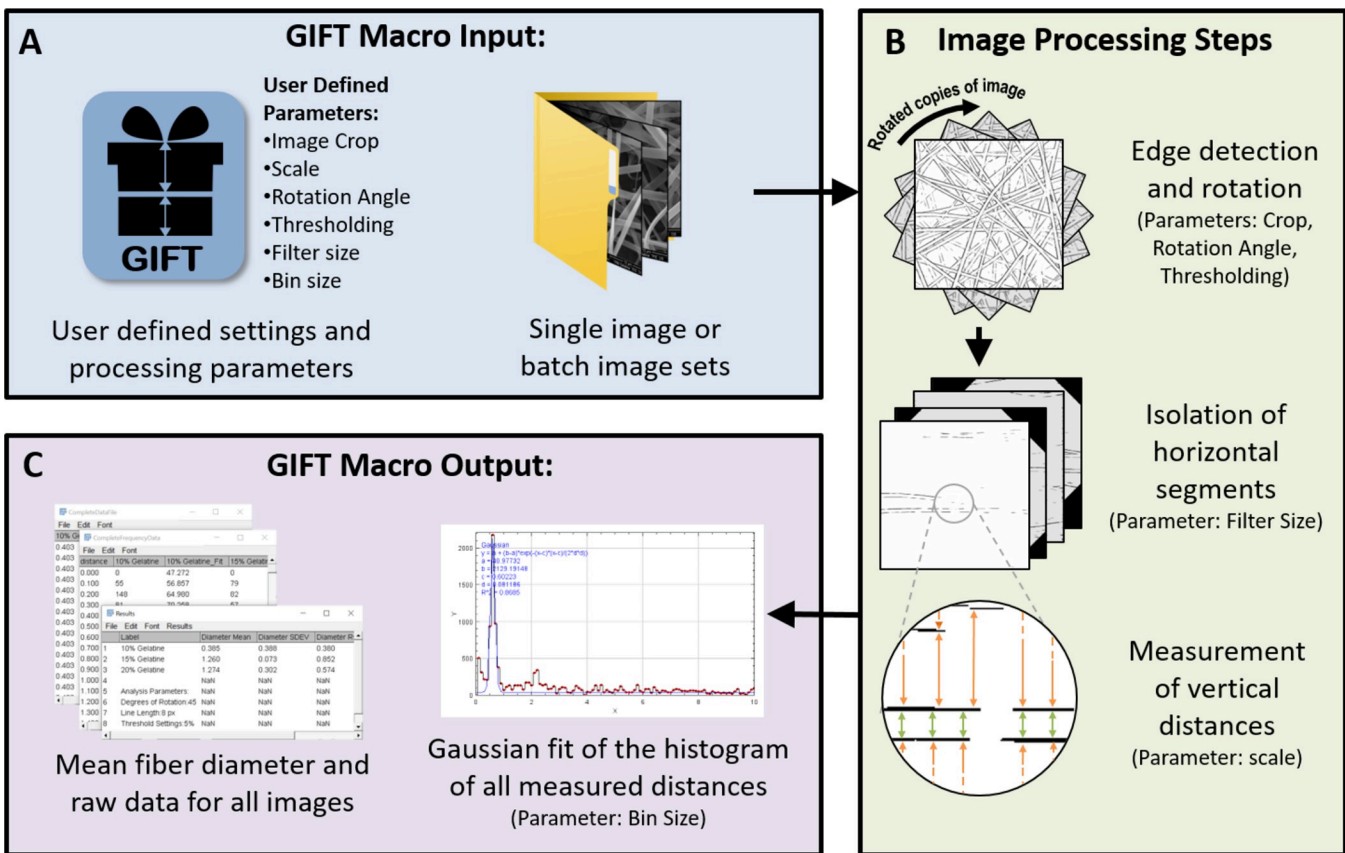

**Fig 1. Diagram of image processing pathway of the GIFT macro.** A) User-defined image processing parameters can be modified in the GIFT graphical user interface. These parameters are applied either to a single image or a batch of images in a single folder. B) Edge detection and thresholding is performed and then a set of rotated copies is made. An opening line filter is applied to isolate horizontal edge segments on each rotated image and the vertical distances between horizontal edge segments are measured. In the figure, the green arrows indicate the actual fiber diameter, which appears with a significantly high frequency in the image. Distances between fibers (orange arrows) are randomly distributed. C) Vertical distances are plotted as a histogram and a Gaussian fit is used to determine average fiber diameter. Users are provided with histograms and raw data for each image and a summary table with all average diameters.

produce more consistent results when analyzing image sets with variable contrast or high blurriness (S1 and S2 Figs in S1 File).

From the binary image containing just the fiber edges (and the edges of any elements or objects in the image), a set of rotated copies of this image is created based on a user-specified angle of rotation (Fig 1B). A horizontal opening line filter is then applied to the all rotated copies using the separate ImageJ plug-in, MorphoLibJ [18]. The MorphoLibJ plug-in is automatically included in FIJI and must be installed when using ImageJ, however, the plug-in provided the filtering step that was initially done with the proprietary software Origin during the development of the GIFT method. The filtering operation isolates horizontal edge segments in each of the rotated copies which should include the parallel edges of fibers. In the next step, the macro scans through each pixel column of each rotated copy and identifies the maxima. Each maximum location corresponds to the presence of an edge in the original image. The vertical pixel distance between each maximum is measured and converted into a real distance based on the user-provided scaling factor. A histogram of all measured distances is plotted. Actual fiber diameters occur with significantly higher frequency and can be identified by applying a Gaussian fit to the data. The peak location of the Gaussian fit is used to determine the mean

fiber diameter and the peak width is used to determine the standard deviation of fiber diameter.

The GIFT macro is designed for ease of use and can be run with minimal user input using the supplied default settings. Only the input of the file or file locations are required to run the macro. However, all of the key processing and output parameters can be modified in a single graphical user interface (GUI). Image cropping, scaling and threshold percent can additionally be determined interactively based on the first image in a given data set at the time of image processing. For additional applicability, users can choose to measure fiber orientation by running the existing plug-ins, OrientationJ or Directionality, in parallel with the GIFT macro and the orientation data is incorporated into the GIFT macro data output for streamlined data processing.

### GIFT benchmarking against existing software with synthetic image set

The average error of the GIFT macro was measured using a set of synthetic images which had been created for the validation of the ImageJ plug-in, DiameterJ, and was published online for the purpose of future benchmarking experiments [19]. Average errors measured with this data set have already been published for the ImageJ-based DiameterJ histogram method and the Matlab-based image analysis tool, SIMPoly [16, 17]. These programs serve as suitable benchmarks as they were specifically designed for fiber measurement in SEM images. Additionally, both programs have varied image analysis methods but utilize fiber diameter histograms to determine the final average fiber diameter in a manner similar to GIFT. Two sets of synthetic test images were used. One composed of straight lines of a constant width spaced at regular intervals, but at variable angles (ordered synthetic images) and the other had randomly placed, curves lines of a constant width (disorder synthetic images). Each set has 3 images each for line widths of 3, 5, 7, 10, 15, 20, 25, 50, 75 and 100 px. Following the precedent set by the SIMPoly analysis, the first 20 px disordered synthetic image was not included due to the extreme incidence of overlapping line segments which made it a poor reference for line width. This image was included in the DiameterJ analysis but did not significantly affect the outcome. Both synthetic image sets were analyzed with the GIFT macro using 8 px filter line length, 1% threshold cutoff and either a 45˚ or 6˚ angle of rotation. The percent error was calculated based on the following equation:

$$\% \ error = \frac{|D_{Real} - D_{Calc}|}{D_{Real}} * 100$$

Where $D_{Real}$ is the known line width in pixels and $D_{calc}$ is the diameter in pixels measured by GIFT. The percent error was calculated for each image and the averages and standard deviations were reported. Average error was calculated separately for 3–7 px lines and 10–100 px lines for better comparison to the previously published error values.

### Electrospun fiber measurement

A series of gelatin electrospun fibers were created as a real-world test set for the GIFT macro. Spinning solutions of 10, 15 or 20% w/v gelatin (Merck, Germany) in 2,2,2-Trifluoroethanol (TFE) were used. Electrospinning was performed at 22˚C and ~24% humidity with a Contipro (Dolní Dobrouč, Czech Republic) 4SpinC4S LAB2 device. Flow rates of 45–130 μl/ml and an applied voltage of 20–30 kV were used to deposit fibers on a cylindrical collector at a distance of 12–60 cm. Parameters were adjusted to ensure reliable fiber formation from each of the different gelatin solutions. Samples were imaged by scanning electron microscopy (SEM) on a Quanta FEG 250 (FEICompany, Germany). Manual measurement of the fiber diameters was

performed by an experienced technician. Ten randomly selected fibers were measured on each image. DiameterJ (version 1.018) was also used to measure the images after cropping and based on the known scale. The fiber diameter histogram mean based on the M3 segmentation method was used for analysis. Because DiameterJ is not compatible with newer versions of ImageJ, this analysis was done using ImageJ 1.52a. SIMPoly was run in Matlab R2020b with the known scale supplied.

The images were batch processed using the GIFT macro using the default image analysis parameters (Line length = 8 px, threshold = 5%, angle of rotation = 45˚), a 0.1 bin size, the scale defined as 32.25 px/µm and the image height cropped to 875 px to remove scale bars.

### GIFT macro parameter testing

The 10, 15 and 20% w/v gelatin electrospun fiber images were analyzed with the GIFT macro under many combinations of image analysis parameters to observe the sensitivity of the results to input parameters. A modified form of the GIFT macro code was used to quickly and automatically run through the various combinations of parameters and record the results. All combinations of filter line length (4–12 px) and threshold percent (2–8%) were tested using 90, 45 or 6˚ angles of rotation for each of the 3 images. These parameter ranges were centered around the default values for the GIFT macro. All analysis was done with a 0.1 bin size and the scale defined as 32.25 px/µm and the image height cropped to 875 px to remove scale bars. The fiber diameter results were graphed as heatmaps using Rstudio. Negative diameter results were graphed as zero. Additionally, two disordered synthetic images from the dataset used from benchmarking (with 10 and 100 pixel diameters) were analyzed with all combinations of 90, 45, 30, 18, 12 and 6˚ angles of rotation and 4–12 px line length. The threshold was kept at 1% because the synthetic images are composed of only black and white pixels.

### Image processing speed measurements

The GIFT macro was tested on a Dell Latitude 5400 (Windows 10 Pro, 8 GB of RAM, Intel Core i5-8365U Processor). The macro was run as part of FIJI, using ImageJ 1.53n. The macro was used to analyze a single image or a set of 10 copies of the 15% Gelatin image shown in Fig 2 (1024x943 pixels) for consistency. Images were analyzed first using the default GIFT settings. To illustrate how some parameters can affect processing time, the same images were also analyzed with the addition of the orientation measurement option (via OrientationJ) or the images were analyzed with a 6˚ angle of rotation instead of 45˚ which increases the number of processed copies of each image from 4 to 30. Each test case was run n = 3 times and mean processing time and standard deviation was calculated.

## Results

### GIFT benchmarking with synthetic images

30 ordered synthetic and 29 disordered synthetic line images from a publically available dataset were analyzed using the GIFT macro [19]. The measured "fiber" diameters were compared to the known diameters and used to calculate average percent error. The percent error was calculated separately for images with fiber widths of 3–7 px and 10–100 px. The percent error generated by GIFT is given in Table 1 alongside the published percent error from measurements on the same dataset using DiameterJ and SIMPoly software. When measuring ordered synthetic images in the 10–100 px range, GIFT run with the default parameter settings had a higher average percent error (6.8±19.0%) than DiameterJ (2.5±1.9%) and SIMPoly (2.1±1.7%). However, the median error value of 1.2% was lower. Additionally, when the angle of rotation was

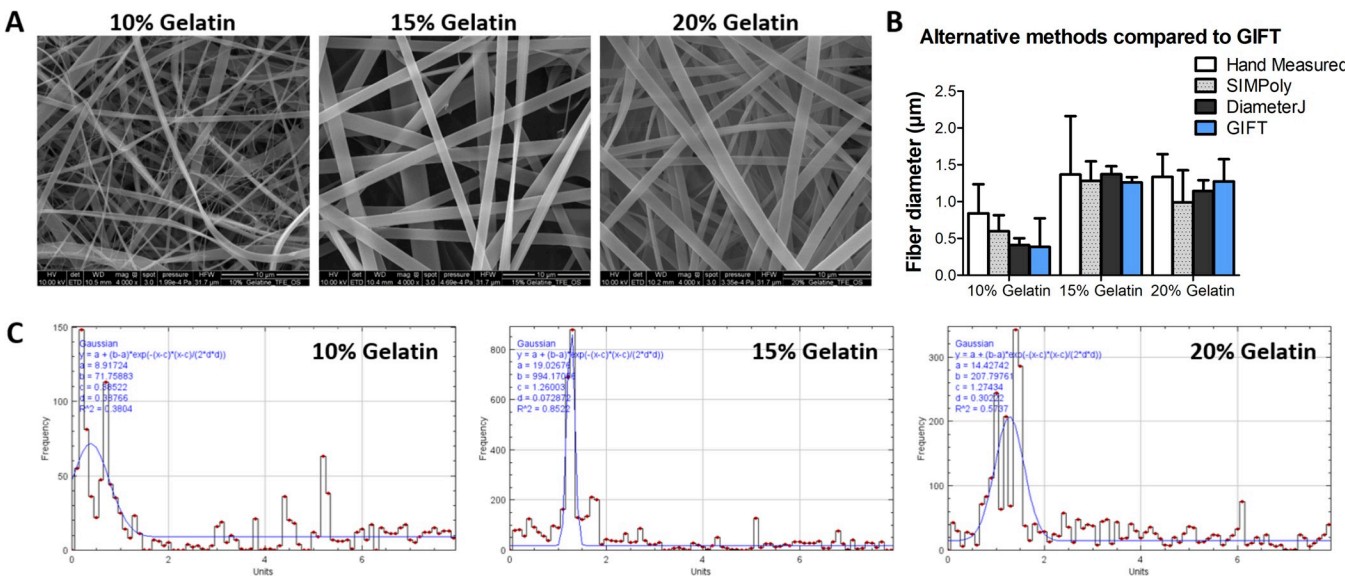

**Fig 2. Real world image analysis.** A) SEM images of electrospun fibers made from 10, 15 or 20% gelatin solutions. Manual fiber diameter measurements were made on 10 random fibers in each image. B) This SEM image set was also used to test the GIFT macro. The images were analyzed using default GIFT parameters. For comparison, the GIFT results were graphed with the results of hand measurement and DiameterJ and SIMPoly analysis of the same images. C) The fiber diameter histograms produced by the GIFT macro for each SEM image are shown. The x-axis units are μm. The blue line overlaid on the histograms represents the Gaussian fit curve. The parameters of the Gaussian fit are shown in the upper left corner.

reduced to 6˚, the average percent error of GIFT (2.2±4.0%) was similar to the other two analysis methods. GIFT performed generally better when analyzing disordered synthetic images, which more closely mimic electrospun fibers. In the 10–100 px range, GIFT run with default parameter settings had an average percent error of 1.2±0.7% which was lower than the error from DiameterJ (4.7±1.4%) and SIMPoly (1.6±1.5%). Lowering the angel of rotation used when analyzing 10–100 px disordered synthetic images did not change the error. When analyzing 3–7 px sized synthetic images, GIFT was only compared to DiameterJ because the values were not published for SIMPoly. For both ordered and disordered synthetic images, GIFT with default settings (ordered: 6.8±6.5%, disordered: 6.0±3.0%) performed significantly better than DiameterJ (ordered: 12.6±3.4%, disordered: 20.9±4.3%).

### Electrospun fiber measurement

10, 15 and 20% gelatin solutions were used to produce electrospun materials which were imaged with SEM (Fig 2A). These images were analyzed by hand and by GIFT. The GIFT macro was run using default parameters and was compared to manual measurements and results from DiameterJ and SIMPoly (Fig 2B). All methods produced similar results, with the

**Table 1. Average percent errors based on analysis of synthetic image set with line widths between 3 and 100 px.**

|  | Line diameter range (pixels) | Ordered Synthetic images | Disordered Synthetic images |
|---|---|---|---|
| **Published DiameterJ average error (%) [19]** | 10–100 | 2.5±1.9 | 4.7±1.4 |
| **Published SIMPoly average error (%) [16]** | 10–100 | 2.1±1.7 | 1.6±1.5 |
| **GIFT average error (%), Default Settings** | 10–100 | 6.8±19.0 (median = 1.2) | 1.2±0.7 |
| **GIFT average error (%), 6˚ angle of rotation** | 10–100 | 2.2±4.0 (median = 1.2) | 1.2±0.7 |
| **Published DiameterJ average error (%) [19]** | 3–7 | 12.6±3.4 | 20.9±4.3 |
| **GIFT average error (%), Default Settings** | 3–7 | 6.8±6.5 (median = 4.6) | 6.0±3.0 |

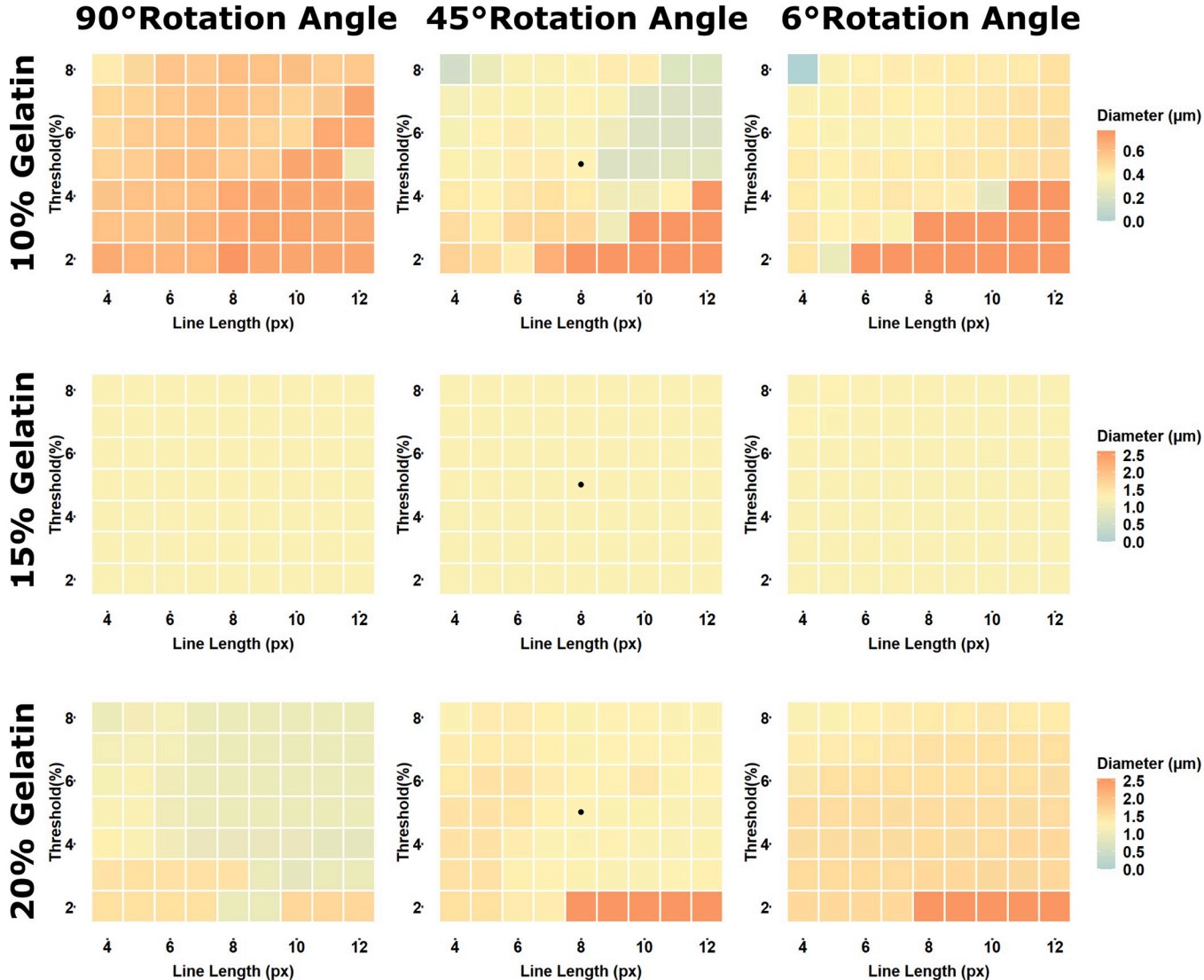

**Fig 3. GIFT parameter sensitivity.** SEM images of 10, 15 or 20% gelatin electrospun fibers were analyzed using the GIFT macro with 90, 45 or 6˚ angles of rotation and at all combinations of 4–12 pixel line length and 2–8% threshold parameters. The resulting fiber diameter measurements were graphed as heatmaps to visualize sensitivity of GIFT to changes in parameters. All graphs for each image (rows) have the same scale, with the fiber diameter measured with default parameters set as the midpoint of the scale bar (yellow color). Average diameters higher than the default are represented by increasingly red hues and diameter results lower than the default are increasingly blue in color. The black dots indicate the results under default parameters for reference.

most noticeable difference in the values for the 10% gelatin image. The resulting fiber diameter histograms for GIFT are given in Fig 2C and show the distributions of measured distances from each image. The blue overlaid line on each histogram shows the Gaussian fit curve used to calculate the average fiber diameter. The Gaussian parameters are given in the top left corner of each graph. For comparison, the fiber diameter histograms for GIFT, SIMPoly and DiameterJ are provided in S5 Fig in S1 File.

## GIFT parameter sensitivity

To assess the sensitivity of the GIFT macro to user input, SEM images of different gelatin electrospun fibers (Fig 2A) were analyzed using the GIFT macro with 90, 45 or 6˚ angles of

**Table 2. GIFT macro processing time under different conditions.**

| | Default settings | Default settings + alignment analysis (OrientationJ) | Minimum angle of rotation (6˚) |
|---|---|---|---|
| **1 Image** | 1.8 ± 0.1 s | 3.3 ± 0.1 s | 21.6 ± 0.9 s |
| **10 Images** | 16.0 ± 0.5 s | 22.6 ± 0.1 s | 215.9 ± 1.0 s |

rotation (which corresponds to 2, 4 or 30 copies analyzed per image, respectively) and at all combinations of 4–12 pixel filter line length and with a 2–8% threshold. The resulting average fiber diameters from each parameter combinations were graphed as heatmaps (Fig 3). Heatmaps for the same image were graphed with a consistent color scale for easier comparison. In all cases, the scales were defined so that the average diameter measured by default parameters represents the midpoint of the scale bar which is set to a yellow color. Average diameters higher than the default are represented by increasingly red hues and diameter results lower than the default are increasingly blue in color. The 15% gelatin image results showed almost no sensitivity to changes in input parameters. The results of the 10 and 20% gelatin images, which represent more difficult images with multiple fibers sizes and lower contrast, respectively, show the most sensitivity to image analysis parameters, but still produce results in a fairly narrow range.

Similar analysis was done for two disorder synthetic images (S3 Fig in S1 File). The results show very little influence of the parameters on the results. With the synthetic images, a percent error can be calculated based on the known line width. The percent error for each set of parameters is shown in S3 Fig in S1 File and also shows very little change in the results based on parameter input.

## Image processing speed measurements

The GIFT macro was tested for processing speed using the 15% gelatin SEM image (Fig 2A). The image was processed once or as a batch of 10 copies and the total processing times are presented in Table 2. The GIFT macro was run using default settings, with the addition of OrientationJ alignment analysis and with a 6˚ angle of rotation for comparison. Under default conditions, 1 image is processed in less than 2 seconds. When using a 6˚ angle of rotation GIFT, which requires processing 30 rotated copies of each image instead of the default 4, the single image processing time was significantly increased (21.6 ± 0.9 s). However, GIFT generally processes images much faster than DiameterJ, which reports a per image processing time of 20 to 60 seconds [17, 19].

## Discussion

The gold standard for average fiber diameter measurement for electrospun materials is to image the material with an SEM and then measure the fiber size by hand, often with the aid of software such as ImageJ. The process of hand measurement is subjective, tedious and does not lend itself to applications which require high throughput analysis like large screening experiments or quality control for product manufacturing. Automated image analysis potentially offers a better alternative to the manual measurement of fibers. However, existing software is either limited by availability, cost, ease of use or accuracy. Our goal was to develop an open source tool that was more accurate than what was currently available and which has a low barrier to entry.

The presented GIFT macro is run through ImageJ, a freely available and popular image analysis software platform which is already familiar to many researchers and has extensive online resources and support. The GIFT macro user manual, available for download at the

Github website along with the program, outlines exactly how to download and open GIFT in ImageJ. To run GIFT, a user only needs to provide an input and output folder location into the single GUI. All other parameters have default values, so the software has a low learning curve. However, all key image processing parameters can be adjusted by the user if desired. Explanations and examples are given in the user manual. The GIFT macro saves image processing intermediates, raw data, fiber diameter histograms and summary data as outputs. Additionally, the GIFT macro has a 'single image' function to allow users to quickly run the analysis method on just one open image.

To assess the accuracy of the GIFT macro in its current form and as a way to benchmark our software with similar tools, we used the GIFT macro to batch analyze a set of synthetic test images specifically designed for validating fiber diameter measurement software. The synthetic test images contained white lines of a constant, known diameter on a black background and were of two types. Ordered images had straight and evenly-spaced lines and disordered images had curved and randomly oriented lines. The average error of the GIFT macro when measuring the line diameter in these images was compared to the previously published average errors for DiameterJ and SIMPoly based on the same dataset [16, 19]. Across the board, GIFT performs similarly or better than DiameterJ and SIMPoly when measuring disordered fibers, which are visibly very similar to the typical structure of electrospun fibers. Importantly, the number of rotations did not significantly affect the error for disorder synthetic images, indicating that, in most cases, there is no need to increase the number of rotated copies per image and subsequently the processing time. In the original DiameterJ paper, line sizes below 10 pixels had an average error over 10% and were not included in the main results, but the values were reported in the supplemental data. GIFT significantly out performs DiameterJ when analyzing the smallest sized lines (3–7 px) which indicates that GIFT may be particularly well-suited to analyzing nanoscale fibers and is not as limited when it comes to minimum measurable fiber size. GIFT had higher average error when analyzing the ordered synthetic images, which are similar in structure to aligned fibers. We believe that one of the strengths of GIFT is that it does not need to specifically identify fibers or background in an image and therefore avoids the need for complicated and potentially inaccurate methods of image segmentation. GIFT relies on the assumption that the segments of the background visible in the image are of random sizes and shapes. However, in the ordered synthetic images both the lines and the background are evenly spaced with parallel edges which means that the background essentially appears like a very wide line and leads to a peak in the fiber diameter histogram generated by GIFT. In these cases, GIFT may identify the repetitive size of the background as the fiber diameter. This did happen in the benchmarking test and impacted the average error for ordered images. However, it happened rarely and the median error was lower than the average error for both DiameterJ and SIMPoly. Additionally, reducing the rotation angle helped to avoid this issue and led to GIFT having an average error similar to the other software. This means that adjusting the rotation angle parameter in GIFT is a simple strategy for the analysis of samples with aligned fibers. Even when an error happens due to highly ordered structures, the actual fiber diameter will still be evident on the histogram, but GIFT currently cannot identify multiple peaks on a single histogram. Luckily, such strictly ordered structures are not typical in electrospun materials.

To test the GIFT macro on real-world images, a set of SEM images of gelatin electrospun fibers was chosen. We felt that the chosen SEM images represented a range of image types and situations that commonly occur. The 15% gelatin image represents an ideal SEM image of fibers. The image is clear, has high contrast and is composed of straight, consistently sized fibers. The 20% gelatin image is similar, but has a lower overall contrast. The 10% image contains a wide range of fiber sizes with the smallest fibers only being a few pixels wide. Therefore

in analyzing just a few images we can demonstrate and assess GIFT's function in a variety of potentially challenging, but common situations. When compared to hand measurements of the same images, GIFT produced averages that were nearly identical. The average diameter measured by the GIFT macro for the 10% gelatin image was slightly lower than the value generated by hand measurement. This may be due in part to the presence of many very thin fibers, which are difficult to measure by hand and easily overlooked during manual measurement analysis. The diameter averages from GIFT were also comparable to the results generated by DiameterJ for the same image set. For some images, GIFT had slightly larger error measurements, however, we believe that this accurately reflects the characteristics of the fibers and that the larger standard deviation corresponding with a wider distribution of fiber sizes. This information can be useful in identifying electrospinning methods that require further optimization to achieve a more consistent fiber size. SIMPoly performed similarly to GIFT in the ideal case of the 15% gelatin image. However, SIMPoly produced different results when challenged with the extremely small fibers or poor contrast in the 10 and 20% images, respectively. The exact histograms generated by each program shown in S5 Fig in S1 File.

DiameterJ and SIMPoly also differ from GIFT in terms of dependence on user input, ease of use and accessibility. DiameterJ requires the selection of a segmentation method based on visual inspection of initial results and this segmentation method can significantly impacts the results (segmentation images shown in S4 Fig in S1 File and diameter results for all segmentation methods included in S1 Dataset). Furthermore, DiameterJ is no longer compatible with current versions of ImageJ which limits its accessibility and utility. SIMPoly runs through Matlab which is not freely available. While SIMPoly is not prone to user bias, it also has very little opportunity for user input and user control which may make it less flexible.

The effects of adjusting image analysis parameters was also evaluated. The GIFT macro has a set of defined default values for all parameters, but they are fully adjustable by the user. The heatmap plots in Fig 3 show how much influence the three main image processing parameters (filter line length, threshold percent and rotation angle), have on the results. In the case of the 15% gelatin image, the parameters have almost no effect on the outcome. Only extreme parameter values generate fiber diameter results which vary widely from those generated by the default settings. In this case, which is an image representative of most fiber SEM images, GIFT is extremely consistent, and not prone to user influence. In the other images, which represent more complicated images with a wide variety of fiber sizes and less ideal image contrast, there is more of an effect of the parameters on the results. However, this variability can be minimized by decreasing the rotation angle and using parameter values near the defaults. Even in these cases the effect on the final fiber size is minimal.

One major current limitation of the GIFT macro is that it is designed for analyzing nonwoven materials with single fiber diameters. The Gaussian fit used to identify the mean diameter from the fiber diameter histogram treats all distributions as if they have one peak. If applied to a bimodal distribution, the Gaussian distribution will only identify one peak or may find the average of 2 close peaks. An example of this can be seen in the fiber diameter histogram of the 10% gelatin fibers. It appears that there is not an even distribution of fiber sizes, but two peaks, one under and one just over 0.5 μm diameter and the Gaussian fit identifies an average between these two close peaks. This average is still a reasonable representation of fiber size, but it means that, in its current form, GIFT may not be ideal for nonwovens with two or more distinct fiber diameters like composite materials made with multiple nozzle or polymers. This limitation is largely because ImageJ is primarily designed for image manipulation and analysis and is not the ideal software environment for doing the more complicated data analysis necessary for analyzing multimodal distributions. That kind of analysis is better done in software that is specifically for statistical analysis. For this reason GIFT saves raw data and binned

histogram data which could be transferred to other software for further analysis if desired. It should be noted that DiameterJ showed multi-peak data in their validation paper, but the multi-peak fitting was done in the separate paid software Igor-Pro. It may be possible in the future to integrate GIFT with external open-source statistical software or to have parallel analysis software developed to be specifically compatible with GIFT output for further analysis. However, this would more than likely not be as user friendly and have a much higher learning curve.

Additionally, GIFT is currently limited to only directly measuring fiber diameters. No other characteristic of the fibers or image is assessed. The GIFT macro can indirectly measure fiber orientation by running the existing plugins Directionality or OrientationJ in parallel to fiber size analysis, but this is a completely separate operation and adds a significant amount of processing time. One of the strengths of GIFT is that it has a very simple and streamlined method for fiber diameter measurement, however, this limits the information that can be mined from GIFTs raw data. There is currently no differentiation between fiber and background which could be used to identify things like pore size. DiameterJ can provide information related to fiber orientation, fiber length, intersection density and mesh hole area, but many of these measurements are based on incorporation of other existing plugins. It may be possible in the future to continue to incorporate more functionality with the GIFT macro by similarly integrating already existing methods.

## Conclusions

The newly developed GIFT macro for ImageJ is a freely accessible and user friendly tool for determining fiber diameter from SEM images of nonwoven biomaterials. The new macro allows users to apply the previously validated GIFT method to large batches of images in a single-step process and has been benchmarked against existing software and tested on a real-word data set. Utilizing this novel method, the GIFT macro determines fiber diameters based on edge distance frequencies and outputs the mean fiber diameter in the image along with standard deviation. Additionally, all raw distance and frequency data is provided to the user for maximum transparency and flexibility. The GIFT macro was designed to maximize user control over image processing and output parameters, but users can start immediately with default settings. Examples and explanations of all image processing steps and parameters are included in the GIFT macro user guide. The macro integrates the existing popular plug-ins OrientationJ and Directionality so fiber orientation data can be collected at the same time as fiber diameter measurements. We believe that the GIFT macro is an easy to use tool with a low barrier to entry that produces reliable fiber diameter measurements and that this tool will be of great interest to researcher using electrospinning or similar techniques for the production of biomaterials.

## Supporting information

**S1 File. Supporting information for the manuscript.**
(DOCX)

**S1 Dataset. Raw data related to the manuscript.**
(XLSX)

## Acknowledgments

The authors would like to thank Katja Hahn and Beate Lyko for their technical assistance and contributions.

## Author Contributions

**Conceptualization:** Jennifer Huling, Andreas Götz, Sabine Illner.

**Data curation:** Jennifer Huling.

**Formal analysis:** Jennifer Huling.

**Software:** Jennifer Huling, Andreas Götz.

**Supervision:** Niels Grabow, Sabine Illner.

**Validation:** Andreas Götz.

**Visualization:** Jennifer Huling.

**Writing – original draft:** Jennifer Huling.

**Writing – review & editing:** Jennifer Huling, Andreas Götz, Niels Grabow, Sabine Illner.

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
