## [Decision Letter · Decision Letter 0]

19 Aug 2022

PONE-D-22-20577GIFT: an ImageJ macro for automated fiber diameter quantificationPLOS ONE

Dear Dr. Jennifer Huling,

Thank you for submitting your manuscript to PLOS ONE. After careful consideration, we feel that it has merit but does not fully meet PLOS ONE’s publication criteria as it currently stands. Therefore, we invite you to submit a revised version of the manuscript that addresses the points raised during the review process. Reviewer 1

Authors introduced the modified GIFT macro program specifically analyzing the diameters of electrospun fibers in SEM images. From my point of view, it would be a very useful tool for researchers doing electrospinning works. Hope to see the further updating functions in the next stage. I only have some minor comments as follows: 1. Maybe details of upgrade information can be listed as a table.

2. Can authors add the produced results of other existing programs as well for the real test case?

Reviewer 2

Author developed a GIFT program that could add onto ImageJ software to help researchers to do automatic nanofiber diameter analysis. However, there are some places author needs to spend more effort so that the GIFT program could be more useful.

First, although GIFT program showed OK performance on current set of SEM images but based on the results and how the image is processed, it looks like this program will have a very hard time to analyze aligned nanofiber. In this situation, a lot of nanofibers will be heavily overlapped with each other. As a big part of nanofiber research field aligned nanofiber shouldn't be ignored.

Second, when measure fiber diameter with the GIFT program, images will be processed by using variable contrast, it may be good for a clear edge (or smooth) nanofiber, but nowadays most of the study will not use plain nanofiber. Researchers will do all kinds of surface treatments to give the fiber more function. The GIFT program should put more effort on how these surface details that change the diameter could be captured.

Last but not the least, deposit a layer of Gold or carbon may help the image qualify but some times to keep the very small details of the nanofiber topography they may not be suitable to be coated before running SEM. In this case the image quality may not be as good. Blur image may also be very normal if the nanofiber is non conductive and thick. Author claimed 10% Gelatin GIFT is more accurate than hand measure I highly doubt that. Due to the lower contrast GIFT got a hard time analyze the fiber underneath on the other hand because nanofiber mesh is a multi-layered structure, researchers will just measure the top layer fibers instead, and take average from more than 5 fields then do statistical analysis.

We look forward to receiving your revised manuscript.

Kind regards,

Wenguo Cui, Ph.D

Academic Editor

PLOS ONE

Journal Requirements:

Reviewers' comments:

Reviewer's Responses to Questions

**Comments to the Author**

1. Is the manuscript technically sound, and do the data support the conclusions?

Reviewer #1: Yes

Reviewer #2: Partly

2. Has the statistical analysis been performed appropriately and rigorously? 

Reviewer #1: Yes

Reviewer #2: Yes

3. Have the authors made all data underlying the findings in their manuscript fully available?

Reviewer #1: Yes

Reviewer #2: Yes

4. Is the manuscript presented in an intelligible fashion and written in standard English?

Reviewer #1: Yes

Reviewer #2: Yes

5. Review Comments to the Author

Reviewer #1: Authors introduced the modified GIFT macro program specifically analyzing the diameters of electrospun fibers in SEM images. From my point of view, it would be a very useful tool for researchers doing electrospinning works. Hope to see the further updating functions in the next stage. I only have some minor comments as follows:

1 Maybe details of upgrade information can be listed as a table.

2 Can authors add the produced results of other existing programs as well for the real test case?

Reviewer #2: Author developed a GIFT program that could add onto ImageJ software to help researchers to do automatic nanofiber diameter analysis. However, there are some places author needs to spend more effort so that the GIFT program could be more useful.

First, although GIFT program showed OK performance on current set of SEM images but based on the results and how the image is processed, it looks like this program will have a very hard time to analyze aligned nanofiber. In this situation, a lot of nanofibers will be heavily overlapped with each other. As a big part of nanofiber research field aligned nanofiber shouldn't be ignored.

Second, when measure fiber diameter with the GIFT program, images will be processed by using variable contrast, it may be good for a clear edge (or smooth) nanofiber, but nowadays most of the study will not use plain nanofiber. Researchers will do all kinds of surface treatments to give the fiber more function. The GIFT program should put more effort on how these surface details that change the diameter could be captured.

Last but not the least, deposit a layer of Gold or carbon may help the image qualify but some times to keep the very small details of the nanofiber topography they may not be suitable to be coated before running SEM. In this case the image quality may not be as good. Blur image may also be very normal if the nanofiber is non conductive and thick. Author claimed 10% Gelatin GIFT is more accurate than hand measure I highly doubt that. Due to the lower contrast GIFT got a hard time analyze the fiber underneath on the other hand because nanofiber mesh is a multi-layered structure, researchers will just measure the top layer fibers instead, and take average from more than 5 fields then do statistical analysis.

6. PLOS authors have the option to publish the peer review history of their article (what does this mean?). If published, this will include your full peer review and any attached files.

Reviewer #1: No

Reviewer #2: No

---

## [Author Response · Author response to Decision Letter 0]

5 Sep 2022

The authors would like to thank both reviewers for their time and helpful feedback. The reviewer’s comments have been formatted into a list and are addressed individually (responses in red). Any changes to the text in response to the reviewer comments are noted in our responses and the changes are marked in the uploaded versions of the manuscript and supporting information with tracked changes. 

Reviewer 1

Authors introduced the modified GIFT macro program specifically analyzing the diameters of electrospun fibers in SEM images. From my point of view, it would be a very useful tool for researchers doing electrospinning works. Hope to see the further updating functions in the next stage. I only have some minor comments as follows:

1. Maybe details of upgrade information can be listed as a table.

Thank you for the suggestion. The underlying code for the GIFT macro was almost completely rewritten to adapt the GIFT method so that it could entirely run in ImageJ. However, there are some key differences in how the analysis was done and major additions to the program in terms of user interactions. We have summarized these main upgrades in a new table in the supporting information (S1 Table in S1 File). 

2. Can authors add the produced results of other existing programs as well for the real test case?

SIMpoly is based in Matlab which means that it is not freely available, but we have used it to analysis our real world SEM image set. DiameterJ is freely available and we have used it to measure the fiber diameters of real world SEM image set. However, DiameterJ is not compatible with the current versions of ImageJ and Fiji. We had to use an older release of ImageJ to run the software. While we think that adding the DiameterJ and SIMPoly results is a useful addition to the paper for further benchmarking of GIFT, we also think that the difficulty in running DiameterJ and accessing SIMPoly further underscores the need for easily accessible software for fiber diameter measurement. The DiameterJ and SIMPoly results have been added to Figure 2B and references to the new data have been added to the methods, results and discussion sections accordingly. Additionally, discussion of the limitation of DiameterJ and SIMPoly has been added to the discussion. 

Reviewer 2

Author developed a GIFT program that could add onto ImageJ software to help researchers to do automatic nanofiber diameter analysis. However, there are some places author needs to spend more effort so that the GIFT program could be more useful.

1. First, although GIFT program showed OK performance on current set of SEM images but based on the results and how the image is processed, it looks like this program will have a very hard time to analyze aligned nanofiber. In this situation, a lot of nanofibers will be heavily overlapped with each other. As a big part of nanofiber research field aligned nanofiber shouldn't be ignored.

It is true that we do not directly measure aligned electrospun fibers, but we do test GIFT extensively using a set of ordered synthetic images which are very similar in structure to aligned fibers. The error produced by GIFT is somewhat higher for these images, but comparable to the other programs tested. We have a thorough discussion of these results in the discussion section. However, it may not have been entirely clear that ordered synthetic images are analogous to aligned fibers, so we have added a statement to this portion of the discussion to make that clearer. We also demonstrate with the ordered synthetic image analysis that decreasing the rotation angle helps to improve the accuracy of GIFT when analyzing aligned structures and we have added discussion about how decreasing the rotation angle parameter in GIFT can be a good strategy for improved accuracy in aligned fiber images. By lowering the rotation angle and therefor creating more rotated image copies during analysis, more fiber edges will be captured during the morphological filtering step. This effect can be seen in the figure below. The same aligned fiber figure was processed with the default settings and settings that are theoretically optimal for aligned and partially overlapping fibers and the morphologically filtered rotated copies are shown. With the decreased angle, there are many more images analyzed and many more fiber edges captured and measured. However, the end result is not remarkably different from the default setting, supporting the fact that GIFT is robust even in the face of aligned fiber images. 

Unfortunately, heavily overlapped fibers are not a problem that GIFT can solve. But neither could a human operator. If one edge of a fiber is completely blocked from view, there is no way to measure it because that information simply is not contained in the image. But as long as only part of a fiber is overlapped, or it is only a few such fibers in an image, GIFT should still perform normally. 

2. Second, when measure fiber diameter with the GIFT program, images will be processed by using variable contrast, it may be good for a clear edge (or smooth) nanofiber, but nowadays most of the study will not use plain nanofiber. Researchers will do all kinds of surface treatments to give the fiber more function. The GIFT program should put more effort on how these surface details that change the diameter could be captured. 

Blurriness at the edges of fibers is discussed more in depth in response to the next comment. We have added new data showing that GIFT can handle some blur in an image and still give accurate results. With regard to surface modification, if a modification changes the fiber diameter, then that will be reflected in the average diameter measured by GIFT, assuming that the surface modification is visible in the image. 

3. Last but not the least, deposit a layer of Gold or carbon may help the image qualify but some times to keep the very small details of the nanofiber topography they may not be suitable to be coated before running SEM. In this case the image quality may not be as good. Blur image may also be very normal if the nanofiber is non conductive and thick. 

Slightly blurry edges on images, either unintentionally or due to circumstances, are an expected challenge when analyzing SEMs of electrospun materials. This is a particularly difficult issue because blurriness obscures the edges of the fibers, which makes analysis difficult no only for computers, but also for human operators measuring by hand. However, the GIFT macro, with its updated thresholding method, has been shown to be more robust in the face of blurry images. We have added a new figure to the supporting information section (S2 Fig in S1 File) which demonstrates how well GIFT deals with blurry edges. These new results show that even some incidence of blurriness, either over the entire image or just on background fibers, should not have a relevant impact on the results generated by GIFT. 

4. Author claimed 10% Gelatin GIFT is more accurate than hand measure I highly doubt that. Due to the lower contrast GIFT got a hard time analyze the fiber underneath on the other hand because nanofiber mesh is a multi-layered structure, researchers will just measure the top layer fibers instead, and take average from more than 5 fields then do statistical analysis.

We agree with the reviewer’s point that we cannot definitively claim that the GIFT results for the 10% gelatin are more accurate. Our intent was to discuss a possible explanation and we have rewritten this sentence in the discussion to remove the mention of accuracy.

---

## [Decision Letter · Decision Letter 1]

19 Sep 2022

GIFT: an ImageJ macro for automated fiber diameter quantification

PONE-D-22-20577R1

Dear Dr. Jennifer Huling,

We’re pleased to inform you that your manuscript has been judged scientifically suitable for publication and will be formally accepted for publication once it meets all outstanding technical requirements.

Kind regards,

Wenguo Cui, Ph.D

Academic Editor

PLOS ONE

Additional Editor Comments (optional):

Reviewers' comments:

Reviewer's Responses to Questions

**Comments to the Author**

1. If the authors have adequately addressed your comments raised in a previous round of review and you feel that this manuscript is now acceptable for publication, you may indicate that here to bypass the “Comments to the Author” section, enter your conflict of interest statement in the “Confidential to Editor” section, and submit your "Accept" recommendation.

Reviewer #1: All comments have been addressed

Reviewer #2: All comments have been addressed

2. Is the manuscript technically sound, and do the data support the conclusions?

Reviewer #1: Yes

Reviewer #2: Yes

3. Has the statistical analysis been performed appropriately and rigorously? 

Reviewer #1: Yes

Reviewer #2: Yes

4. Have the authors made all data underlying the findings in their manuscript fully available?

Reviewer #1: Yes

Reviewer #2: Yes

5. Is the manuscript presented in an intelligible fashion and written in standard English?

Reviewer #1: Yes

Reviewer #2: Yes

6. Review Comments to the Author

Reviewer #1: (No Response)

Reviewer #2: Author properly addressed all the comments from my previous review with more description in details. Author also added new figures in supplement material to better demonstrate how GIFT could handle with blurry edges which could ensure the analyzed results be more accurate and reliable. No more new comment at this point. Thank you.

7. PLOS authors have the option to publish the peer review history of their article (what does this mean?). If published, this will include your full peer review and any attached files.

Reviewer #1: No

Reviewer #2: No

---

## [Editor Report · Acceptance letter]

23 Sep 2022

PONE-D-22-20577R1 

GIFT: an ImageJ macro for automated fiber diameter quantification 

Dear Dr. Huling:

I'm pleased to inform you that your manuscript has been deemed suitable for publication in PLOS ONE. Congratulations! Your manuscript is now with our production department. 

Kind regards, 

on behalf of

Professor Wenguo Cui 

Academic Editor

PLOS ONE